# Characterising the Physiological Responses of Chinook Salmon (*Oncorhynchus tshawytscha*) Subjected to Heat and Oxygen Stress

**DOI:** 10.3390/biology12101342

**Published:** 2023-10-17

**Authors:** Roberta Marcoli, Jane E. Symonds, Seumas P. Walker, Christopher N. Battershill, Steve Bird

**Affiliations:** 1Centre for Sustainable Tropical Fisheries, College of Science and Engineering, James Cook University, Townsville, QLD 4811, Australia; roberta.marcoli.95@gmail.com; 2ARC Research Hub for Supercharging Tropical Aquaculture through Genetic Solutions, James Cook University, Townsville, QLD 4811, Australia; 3School of Science, University of Waikato, Private Bag 3105, Hamilton 3240, New Zealand; christopher.battershill@waikato.ac.nz; 4Cawthron Institute, Nelson 7010, New Zealand; jane.symonds@cawthron.org.nz (J.E.S.); seumas.walker@cawthron.org.nz (S.P.W.)

**Keywords:** heat shock protein (HSP), superoxide dismutase (SOD), stress, Chinook salmon, qPCR, climate change

## Abstract

**Simple Summary:**

It is widely accepted that climate change is a rapidly increasing threat to the aquaculture industry, with the biggest issue being a slow but continuous rise in water temperature and an associated decrease in dissolved oxygen in some areas. In New Zealand fish farms, during the hottest period of the year (January/February), increased water temperatures have been shown to be detrimental to Chinook salmon performance and are associated with mortalities. We report on the responses of Chinook salmon to a chronic, long-term increase in temperature and decrease in oxygen over a three-month period. We also identified, characterised, and developed relevant biomarkers for Chinook salmon and applied them to monitor responses to changes in the fish’s environment. This paper provides evidence on how the use of molecular tools, alongside general health and haematological parameters, have the potential to better inform the aquaculture industry of stock health status, thereby enhancing resilience, efficiency, and productivity.

**Abstract:**

In New Zealand, during the hottest periods of the year, some salmon farms in the Marlborough Sounds reach water temperatures above the optimal range for Chinook salmon. High levels of mortality are recorded during these periods, emphasising the importance of understanding thermal stress in this species. In this study, the responses of Chinook salmon (*Oncorhynchus tshawytscha*) to chronic, long-term changes in temperature and dissolved oxygen were investigated. This is a unique investigation due to the duration of the stress events the fish were exposed to. Health and haematological parameters were analysed alongside gene expression results to determine the effects of thermal stress on Chinook salmon. Six copies of heat shock protein 90 (HSP90) were discovered and characterised: HSP90AA1.1a, HSP90AA1.2a, HSP90AA1.1b, HSP90AA1.2b, HSP90AB1a and HSP90AB1b, as well as two copies of SOD1, named SOD1a and SOD1b. The amino acid sequences contained features similar to those found in other vertebrate HSP90 and SOD1 sequences, and the phylogenetic tree and synteny analysis provided conclusive evidence of their relationship to other vertebrate HSP90 and SOD1 genes. Primers were designed for qPCR to enable the expression of all copies of HSP90 and SOD1 to be analysed. The expression studies showed that HSP90 and SOD1 were downregulated in the liver and spleen in response to longer term exposure to high temperatures and lower dissolved oxygen. HSP90 was also downregulated in the gill; however, the results for SOD1 expression in the gill were not conclusive. This study provides important insights into the physiological and genetic responses of Chinook salmon to temperature and oxygen stress, which are critical for developing sustainable fish aquaculture in an era of changing global climates.

## 1. Introduction

Globally, marine environmental temperatures are increasing, and the combination of thermal stress and potential hypoxia are creating an environmental challenge for the aquaculture industry [1,2,3]. It is widely accepted that ocean temperatures have increased over the last few decades and this is predicted to continue [4]. This is noticeable during summer periods within New Zealand, where record temperatures are being reached [5]. The long-term trend is towards warmer oceans around New Zealand and in the Tasman Sea, depending on the location and season [6]. The optimal temperature for New Zealand Chinook salmon (*Oncorhynchus tshawytscha*) is between 15–17 °C, in which the best health and growth are achieved [7]. However, within the Marlborough Sounds, where many Chinook salmon are currently farmed, the last five years have seen peak summertime water temperatures exceed 18 °C at some farm sites and increased mortalities have been reported [8,9]. Higher mortality correlated with heat waves and chronic thermal stress is not unexpected and has been observed in other salmonid species, such as rainbow trout, where high temperatures are considered the main reason for a mortality syndrome during summer [10]. Fish can adapt and cope with daily and seasonal water temperature variations [11], but due to accelerated global warming, farmed poikilothermic fish, such as salmonids, could be under threat.

A number of markers are available to measure stress events in fish. Changes in the blood plasma and haematology have been utilised to describe haematological changes [12,13,14,15,16,17,18]. Cortisol, which is produced under nervous system control by the hypothalamus–pituitary–interrenal pathway, has also been widely used [19,20,21,22,23]. Additionally, lactate levels have been monitored [24,25,26], where anaerobic metabolism and lactate accumulation have been correlated with anaerobic fuel supply in fish subjected to environmental and functional hypoxia accumulation [27,28,29,30]. However, more recently, with the increased availability of genomic information for multiple fish species, researchers are using specific genes involved in multiple biological pathways as biomarkers to understand physiological changes due to stress [31,32]. The heat shock proteins (HSP) are a family of proteins, some of which are produced when cells are exposed to temperatures outside their optimal range [33,34,35]. This is a universal response, found in virtually every living organism, including bacteria and plants [36]. HSPs perform chaperone functions, where they stabilise newly synthesised proteins to ensure they fold correctly, help to refold proteins damaged by stress, or degrade unrepairable damaged proteins [37,38]. It has been widely shown that the expression of HSP90 changes in response to a variety of stress factors, including temperature changes [39] and decreased oxygen [40]. HSP90 has previously been utilised as a biomarker of thermal stress in fish [41,42,43,44] particularly in salmonid studies [45,46,47,48]. To date, six copies of the HSP90 gene have been characterised within Atlantic salmon and rainbow trout [49,50], whereas an in-depth characterisation of these genes has not been carried out in Chinook salmon, with only some copies used as biomarkers of stress [20,51]. Another marker of temperature stress is superoxide dismutase (SOD), whose role is to buffer the production of superoxide, one of the main reactive oxygen species (ROS) generated within the cell [52]. Several studies in fish have shown that changes in temperature have a noticeable effect on oxidative stress [53,54,55,56,57,58,59], and, more specifically, on the expression of SOD1, which has already been utilised as a stress biomarker in different fish species, including salmonids [60,61,62,63,64]. However, a full characterisation of SOD1 in salmonids, including Chinook salmon, remains to be carried out.

This study investigated Chinook salmon physiological responses when exposed to a long-term (up to 71 days) high temperature and reduced dissolved oxygen (DO) challenge in a recirculating aquaculture system (RAS). To understand the effects of this challenge, plasma biochemistry (cortisol and lactate), haematology and performance parameters (growth, mortality, feed intake and organ indices) were analysed in a subset of fish sampled from this trial. In addition, using the Chinook salmon genome, all copies of the housekeeping genes, RNA Polymerase II, I and III Subunit F (POLR2F) and Ribosomal Protein L18 (RPL18) and stress genes, HSP90 and SOD1 were identified, and their expression patterns investigated using qPCR in the liver, spleen, and gills. Our results provide important insights into the overall effects that chronic thermal and reduced oxygen stress have on Chinook salmon, through the use of specific stress biomarkers. Understanding the metabolic status and health of fish during chronic thermal stress, will enable the aquaculture industry to make better-informed management decisions during these events.

## 2. Materials and Methods

### 2.1. Chinook Salmon HSP90, SOD1, POLR2F and RPL18 Gene Discovery and Synteny Analysis

Chinook salmon HSP90, SOD1, POLR2F and RPL18 gene sequences were searched for within the available genome [65] and the conservation of synteny was investigated between selected vertebrate genomes and the Chinook salmon genome (Otsh_v1.0, GenBank assembly accession: GCA_002872995.1; Otsh_v2.0, GenBank assembly accession: GCA_018296145.1). Initially, the basic local alignment search tool (BLAST) [66] was used to search the Chinook salmon genome database using known rainbow trout HSP90, SOD1, POLR2F and RPL18 genes. Homologues for each gene were identified within different DNA chromosomes or scaffolds found within the databases. Subsequently, GENSCAN [67], Fgenesh [68], BLAST [66], FASTA [69] and the Translate tool in ExPASy [70] were used to annotate the genes of interest and other genes found within their vicinity on each chromosome or scaffold. Using the above approach, the genomes of selected vertebrate species (Table 1) found in Ensembl [71] were also analysed to determine the level of synteny conserved between species around the HSP90, SOD1, POLR2F and RPL18 genes. From the Chinook salmon genome, gene organisation and predicted amino acid sequences for all genes used in this investigation were determined for comparison between species.

### 2.2. Sequence Alignments and Phylogenetic Analysis

Selected vertebrate HSP90, SOD1, POLR2F and RPL18 sequences were retrieved from the NCBI database. Multiple sequence alignments of selected amino acid and nucleotide sequences were generated using Clustal omega [72] and phylogenetic relationships were constructed from the alignments of the full-length amino acid sequences of known vertebrate HSP90, SOD1, POLR2F and RPL18 molecules using the neighbour-joining method [73]. Each tree generated was edited using the Interactive Tree Of Life (iTOL) v6.6 [74] or TreeView [75] and confidence limits were added [76]. Finally, the predicted amino acid sequences were analysed using SignalP v6.0 [77] and important protein structures were predicted using ScanProsite [78].

### 2.3. Fish Rearing Prior to the Temperature/DO Trial

All female Chinook salmon were sourced from two commercial freshwater hatcheries. The trial fish were transferred to the Finfish Research Centre (FRC) at the Cawthron Aquaculture Park, Glenduan, New Zealand on the 17th and 18th December 2018. A total of 4402 juveniles, individually tagged with Passive Integrated Transponder (PIT) tags (HIDGlobal, EM4305, 12 mm long and 2 mm diameter glass tags) were transferred and, on arrival, were split into eight 8000 L tanks with salinity at 18–20 ppt and a temperature of 15 °C. The temperature began to be increased by 0.5 °C per day up to 17 °C starting on the 20th December 2018, and fish were acclimatised to full seawater (35 ppt) over a period of 16 days. From December 2018 to November 2019 the fish were part of a separate study on feed intake and growth. During this period, all trial tanks received recirculated and disinfected oxygenated seawater at 17 °C. The photoperiod was set as 24 h artificial light and fish were hand fed to satiation on a commercial diet. Pellet size (4, 6, and 9 mm) was increased as the fish grew, following the manufacturer’s guidelines. Fish numbers were reduced during the trial to keep densities below ~25 kg/m^3^. After one year of rearing (from December 2018 to November 2019), the fish with poor performance or with deformities were euthanised and 329 fish (weight 2103 g ± 443 g and mean fork length 451 mm ± 26 mm) were kept and reared in 12 × 8000 L tanks for the trial in this investigation.

### 2.4. FRC Rearing Systems

The FRC rearing systems used for the high-temperature/low-DO challenge trial (December 2019 to February 2020) consisted of three recirculating aquaculture systems (RAS), each with four or five tanks, as described previously [79]. All three systems were supplied with disinfected seawater from the same intake. Each tank contained 8000 L of water, which was replaced twice per hour (150 L/min). The outflowing water from each system was collected in a sump, and subsequently underwent six treatment steps (drum filtration, biofilter, foam fractionator injected with ozone, heat pump for temperature control, UV disinfection and oxygen injection) to ensure sufficient treatment and reoxygenation of at least 90% of the wastewater (Appendix A).

### 2.5. Chinook Salmon Temperature/DO Trial

#### 2.5.1. Fish Husbandry during the Trial

During the high-temperature/low-DO challenge trial, fish were hand fed to satiation one meal of 9 mm pellets each morning. Approximately 15 min after each meal, the uneaten feed was recovered from each tank separately and then dried before being counted to determine the amount of uneaten feed and the total feed consumed by the fish in the tank. Fish were closely observed during feeding and regularly checked during the day, with all mortalities removed from the tanks, recorded, and weighed.

#### 2.5.2. Assessment of Fish for the Trial

The trial fish were assessed from 18 November 2019 to 3 December 2019, and a sub-set were sampled before the high-temperature/low-DO challenge trial began (pre-trial group). Prior to each assessment, fish were crowded in the rearing tank and anaesthetised in bins containing 65 mg/L of tricaine methanesulfonate and oxygenated seawater. Tagged fish were scanned into the computer using an Avid-Power Tracker VI microchip reader (Avid Identification Systems, Norco, CA, USA) before the measurements were carried out. All fish were weighed using a digital balance. The fork length and girth of each fish were measured to the nearest mm using a ruler and tape measure, respectively. External appearance was assessed and only fish that appeared healthy, had gained weight, and were in good condition were kept for the high-temperature/low-DO trial. A total of 329 individuals were kept, whose mean weight was 2103 g ± 443 g. Detailed necropsies were carried out on 148 fish at the start of the trial, which included internal inspection and organ weights (heart, liver, gonad). At the end of the high-temperature/low-DO trial in February 2020, all fish (*n* = 84 at 19 °C and *n* = 22 at 17 °C) underwent necropsies, and 16 fish from the control tanks and 16 from the high-temperature/low-DO tanks had their tissues sampled for further analysis. The performance and health data in this study are based on the subset of fish that underwent in-depth sampling (*n* = 48). Results from the larger study have previously been published [80]. The specific growth rate (*SGR*) was calculated using the following formula, where 
$w_{f}$
 = final weight in grams and 
$w_{i}$
 = initial weight in grams:
$$SGR=\frac{\ln\left(w_{f}\right)-\ln\left(w_{i}\right)}{days}\times 100$$


#### 2.5.3. Trial Set-Up

Three RAS systems were used for the trial: RAS systems A and C for the high-temperature and low-DO challenge, and RAS system B for the control (Figure 1). To allow for staggered sampling at the end of the trial, when only two tanks per day could be assessed and sampled effectively, the temperature increases for system A and C tanks commenced on different dates. For the control fish, two tanks were kept at a constant 17 °C, with DO levels set to between 7.0 and 8.0 mg/L during the trial. For the temperature challenge, fish were reared in ten tanks, 5 tanks in RAS system A and 5 in RAS system C. Fish numbers per tank ranged from 18 to 36, with a density not exceeding 25 kg/m^3^. On commencement of the temperature challenge, the temperature was slowly increased over 12 days by 0.25 °C per day over a 5 h period, from 17 °C to 20 °C. In addition to temperature changes, DO levels were also decreased in the tanks in RAS system A and C, attempting to mimic the DO levels found over summer in marine farms. Two days prior to the start of the temperature increase, the amount of supplemental oxygen added to the tanks was reduced to target DO levels between 5.5 and 6.5 mg/L. The temperature increase in RAS system A commenced on 12 December 2019 and 5 days later it was commenced in RAS system C. Overall, the number of days the fish spent at an elevated temperature and lowered DO was 68 to 70 days in RAS system A and 66 to 71 days in RAS system C.

#### 2.5.4. Trial Adjustments

To ensure that fish survived until the end of the trial to assess health and growth, conditions were adjusted as the trial progressed (Figure 1). In RAS system A, after four days at 20.0 °C, feed intake was reduced significantly, and four mortalities occurred. The temperature was decreased to 19.5 °C, with a similar decrease subsequently applied to RAS system C. The temperature was again decreased by 0.5 °C to 19.0 °C on the 30 January 2020 in both systems. During the trial, DO levels were also increased in mid-January to between 6.5 to 7.0 mg/L and again increased on 30 January 2020 to a target DO of between 7.0 to 7.5 mg/L, for the same reasons as the temperature changes. This temperature and DO were then maintained in the ten challenge tanks until the end of the trial, when all fish were euthanised between 17–19 February 2020 (RAS system A), 24–25 February 2020 (RAS system C) and 27–28 February 2020 (RAS system B).

### 2.6. Fish Assessment and Sampling

Blood samples for plasma haematology were collected (without anti-coagulants) from the caudal vein immediately following euthanasia from a sub-set of fish from each tank pre-trial (*n* = 16) and end of the trial (*n* = 32). Fish were euthanised using an overdose of 100 ppm AQUI-S^®^ (AQUI-S New Zealand Ltd., Lower Hutt, New Zealand) in seawater. For all fish, biometrics such as weight, length, and girth were measured and liver, spleen, and gills were sampled from 16 fish taken from the control-temperature group (8 fish per tank) and from 16 fish taken from the high-temperature and low-DO group (4 fish per tank, two tanks from RAS A and from RAS C). These tissues were placed in RNAlater™ Stabilization Solution (Thermo Fisher, Waltham, MA, USA) and stored at −80 °C prior to RNA extraction. Blood samples were processed and analysed according to standardised methods [81]. In brief, fresh peripheral blood samples were centrifuged at room temperature (16,250 rcf for 8 min) to obtain plasma, transferred to 2 mL sterile cryovials (Interlab; Auckland, New Zealand) and immediately snap-frozen in liquid nitrogen. Subsamples were sent on dry ice to Gribbles Veterinary Laboratory, New Zealand for targeted and quantitative analyses of selected biochemical and haematology parameters. Plasma cortisol and lactate levels (nmol/L) were determined using an automated endocrinology Cobas e 411 analyser (Roche Diagnostics, Mannheim, Germany). Each assay used a standard kit (Roche Diagnostics, Mannheim, Germany) developed for the automated analyser. Haematology analysis was conducted according to previously described methods [81] and included haemoglobin (Hb) levels (g/L), haematocrit (Hct) levels (%), white blood cell counts (WBC) and differential leukocyte counts. The % mean corpuscular haemoglobin concentration, (haemoglobin/haematocrit × 100) and the hepatosomatic Index (HSI) (liver weight/body weight × 100) was also calculated.

ANOVA and *t*-tests were performed for the analysis of the health measurements. Before pooling the results from tanks of the same treatment, a chi-square test was performed to make sure there was no statistical difference across replicates.

### 2.7. RNA Extraction and cDNA Synthesis

Total RNA was isolated from ~100 mg of tissue taken from the liver, spleen, and gills using the easy spin [DNA free] Total RNA Extraction Kit (iNTrON Biotechnologies, Republic of Korea), following the manufacturer’s instructions. Briefly, the tissue was homogenised in easy-BLUE reagent (iNTrON Biotechnologies, Gyeonggi-do, Republic of Korea) with silica beads using a Precellys® Evolution Touch Homogenizer (Bertin Technologies, Paris, France). Chloroform was added and the samples were then centrifuged for 10 min at 4 °C, to ensure separation between RNA and the rest of the cell debris. The supernatant liquid, where RNA was present, was then washed and purified by following the column-based kit protocol. The RNA was re-suspended in 50 µL of Elution Buffer and RNA concentration and purity was determined using the Nanodrop DeNovix DS-11 Series Spectrophotometer (Thermofisher, Waltham, MA, USA) and the integrity of the RNA was checked using electrophoresis and imaged using the iBright™ CL750 Imaging System (Thermofisher, Waltham, MA, USA). cDNA was produced from 3 µg of previously extracted total RNA, using the HiSenScript^TM^ RH(-) RT PreMix kit (iNTrON Biotechnologies, Gyeonggi-do, Republic of Korea), following the manufacturer’s instructions. Briefly, RNA was added to each HiSenScript™ RH(-) RT PreMix tubes and Dnase/Rnase-Free water used to give a 20 µL final volume. Tubes were then placed into a SimpliAmp™ Thermal Cycler (Thermofisher, Waltham, MA, USA) and incubated at 45 °C for 1 h, followed by 85 °C for 10 min. After incubation, each cDNA sample was diluted 3 times using Dnase/Rnase-Free water and stored at −20 °C prior to use in quantitative PCR (qPCR).

### 2.8. Primer Design and Quantitative PCR (qPCR)

Using the predicted Chinook salmon gene sequences for SOD1, HSP90, RPL18 and POLR2F, primers were designed for use in qPCR (Table 2).

Due to the high similarity of the multiple gene sequences for each gene, primers for both housekeeping genes and gene of interests were designed to ensure that all copies of each gene were captured in the expression study. At least one of the primers of each pair was designed over an exon–exon boundary, to ensure that no genomic DNA could be amplified and each primer pair tested using cDNA and genomic DNA as a template. qPCR was performed using a 48 well Magnetic Induction Cycler (MIC) qPCR cycler (BMS, Yatala, Australia). For each reaction, 5 µL of cDNA, 1 µL of forward and 1 µL reverse primer, 10 µL of Bio-Rad SsoAdvanced Universal SYBR Green Supermix and 3 µL of DNase/RNase free water was added to a MIC tube (BMS, Yatala, Australia), to give a final reaction volume of 20 µL. Each sample was carried out in duplicate, and a double-negative control was also performed for each primer pair. Amplification was carried out using the following conditions: one cycle of 98 °C for 30 s, 40 cycles of 98 °C for 15 s and then an optimal annealing temperature that had been determined previously (Table 2), for 30 s. The fluorescence signal output was measured at 60 °C for each tube after each cycle and a melting curve for each qPCR was determined at the end of the programme by reading the fluorescence at every degree from 65 °C to 95 °C to ensure that a single product had been amplified. To also check that the correct product had been amplified, each real-time product was run on a 2% agarose gel, stained with RedSafe and viewed under UV light and imaged using the iBright™ CL750 Imaging System (Thermofisher, USA), to confirm the correct band size. Some of these bands were also sent for sequencing to confirm that the correct gene was being amplified. Primer efficiency for each run was obtained using the LinRegPCR algorithm [82] and expression levels of HSP90 and SOD1 were normalised to two house-keeping genes, RPL18 and POLR2F (Table 2). The expression of SOD1 and HSP90 in each tissue analysed was achieved using the geometric means of two housekeeping genes [83], using the Pfaffl method [84] and presented as normalised fold expression [85]. Statistical significance was determined using the F test, and, depending on the *p*-value result, either a *t*-test assuming unequal variance or a *t*-test assuming equal variances was performed, and the *p*-value of the *t*-test was analysed, where values *p* < 0.05 were considered significant.

## 3. Results

### 3.1. Daily Temperature and DO of Each RAS System during the Trial

Temperature and DO were recorded every 15 min and the daily means were reported for each RAS system (A-C) for the duration of the trial (Figure 1A and Table S4 from [80]).

### 3.2. Chinook Salmon Performance Parameters

#### 3.2.1. Fish Weight, Weight Gain and Hepatosomatic Index (HSI) at the End of the Trial (Sampled Population)

The mean weight of fish kept at the high-temperature and low-DO levels and sampled for further analysis was 2505.75 ± 160.49 g, which was significantly lower (*p* < 0.05) than the mean weight of the control group, which was 2954.75 ± 137.57 g (Figure 2A). No statistically significant difference was seen between the mean weight of these fish in the two groups at the start of the trial. Significant (*p* < 0.02) weight loss occurred in the fish kept at the high-temperature and low-DO levels, where the mean weight loss was −38.06 ± 1.15 g, while the mean weight gain in the control tanks was 225.76 ± 2.99 g (Figure 2B). There was a significant decrease (*p* < 0.02) in the HSI of fish in the high-temperature and low-DO tanks (0.98) compared to the control tanks (1.31) (Figure 2C).

#### 3.2.2. Feed Intake Per Fish and Fish Mortalities during the Trial (Whole Population)

Tank feed intake was measured after each meal and analysed as the mean daily feed intake per week, divided by the number of fish in the tank. For the control temperature, the mean feed intake of the two tanks used in RAS system B, showed no decrease over time and the highest food intake was seen at the end of the trial, 97 g per fish per week (Figure 3A). In the challenge tanks, the mean feed intake decreased over time. An average of the ten tanks used in RAS system A and C, showed that the highest food intake was 88.3 g per fish per week at the beginning of the trial and the lowest feed intake was 15.4 g per fish per week towards the end of the trial (Figure 3A). When the whole population of fish sampled was considered (45 fish from the control tanks and 250 fish from the high-temperature/low-DO tanks), there was a significant (*p* < 0.02) decrease in the SGR of fish kept at the high-temperature and low-DO, where the mean SGR was −0.021, compared to the control, which was 0.235.

Cumulative mortality was calculated by summing the total number of mortalities each day over the length of the trial in the control or the high-temperature and low-DO tanks. Total numbers of mortalities were expressed as the percentage of the starting number used in the two treatments. The cumulative fish mortality in the high-temperature and low-DO tanks was 22.0% and in the control temperature tanks was 4.6% (Figure 3B).

### 3.3. Chinook Salmon Haematology and Plasma Biochemistry

The blood parameters analysed in the pre-trial, control and high-temperature and low-DO fish are summarised in Table 3. Statistically significant differences were found in the neutrophils %, lymphocytes % and the neutrophil/lymphocyte ratio between fish kept at high-temperature/low-DO and the fish from the pre-trial control and time-matched control temperatures.

The mean lactate concentration between the pre-trial fish (17.72 mmol/L) and the time-matched control fish (17.64 mmol/L) was not significantly different (Figure 4A). However, the average lactate concentration in the fish kept at high-temperature and low-DO (24.86 mmol/L) was significantly higher than the pre-trial fish (*p* < 0.02) and the time-matched control fish (*p* < 0.02). The average cortisol concentration found between the pre-trial fish (267.96 nmol/L) and the control fish (231.94 nmol/L) was not significantly different (Figure 4B). There was also no significant difference between the average cortisol concentration in the fish kept at the high-temperature and low-DO (341.02 nmol/L) and the pre-trial fish (*p* = 0.09) and the time-matched control fish (*p* = 0.06).

### 3.4. Characterisation of Chinook Salmon Housekeeping Genes

Chinook salmon POLR2Fa (Appendix A; GenBank accession No. OQ215307) and POLR2Fb (Appendix A; GenBank accession No. OQ215308) cDNA coding sequences and genes were predicted using chromosomes LG09 and LG27, respectively. Both cDNA sequences were 384 nucleotides (nt) in length and had a gene organisation of five exons and four introns, with the intron splicing consensus (GT/CAG) conserved at the 5′ and 3′ ends. A high degree of synteny was found between the regions of the Chinook salmon and rainbow trout genomes and human chromosome 22, that contained the POLR2F genes (Appendix A), with MICAL like 1 (MICALL1) and SRY-Box Transcription Factor 10 (SOX10) found either side of the POLR2F genes in all genomes. Both Chinook salmon POLR2F cDNA’s encoded a 127 amino acid sequence, with no signal peptide predicted using SignalP. Alignment of the predicted POLR2F protein sequences with selected known vertebrate POLR2F sequences showed two conserved regions, an acidic segment in the N-terminal region and a C-terminal region, essential for the enzymes’ activity and containing the RNA polymerases K/14 to 18 Kd subunits’ signature, [ST]-x-[FY]-E-x-[AT]-R-x-[LIVM]-[GSA]-x-R-[SA]-x-Q (Appendix A). Additional conserved sites included a Casein kinase II phosphorylation site, cAMP- and cGMP-dependent protein kinase phosphorylation site and Tyrosine kinase phosphorylation site 2. The close relationship of the Chinook salmon POLR2F sequences with other known fish POLR2F sequences, especially rainbow trout, is also apparent in the phylogenetic tree (Appendix A). Due to the high similarity of the Chinook salmon POLR2F sequences, primers were designed to amplify both expressed forms of the gene for qPCR (Appendix A).

Chinook salmon RPL18a (Appendix A; GenBank accession No. OQ215309) and RPL18b (Appendix A; GenBank accession No. OQ215310) cDNA coding sequences and genes were predicted using chromosomes LG03 and LG23, respectively. Both cDNA sequences were 567 nt in length and had a gene organisation of seven exons and six introns, with the intron splicing consensus (GT/CAG) conserved at the 5′ and 3′ ends. A high degree of synteny was found between the regions of the Chinook salmon and rainbow trout genomes and human chromosome 19, that contained the RPL18 genes (Appendix A), with Sphingosine kinase 2 (SPHK2) found next to the RPL18 genes in all the genomes. Both Chinook salmon RPL18 cDNA’s encoded a 188 amino acid sequence, with no signal peptide predicted. Alignment of the predicted RPL18 protein sequences with selected known vertebrate RPL18 sequences, showed conserved regions, including a ribosomal protein L18e signature, [KREWDI]-x-L-x(2)-[PSRG]-[KRS]-x(2)-[RHY]-[PSA]-x-[LIVM]-[NSA]-[LIVM]-x-[RK]-[LIVM], a casein kinase II phosphorylation site, two N-myristoylation sites, two amidation sites, an N-glycosylation site, cAMP-and cGMP-dependent protein kinase phosphorylation site and protein kinase C phosphorylation sites (Appendix A). The close relationship of Chinook salmon RPL18 with other known fish RPL18 sequences, especially rainbow trout, is also apparent in the phylogenetic tree (Appendix A). Due to the high similarity of the Chinook salmon RPL18 sequences, primers were designed to amplify both expressed forms of the genes for qPCR (Appendix A).

### 3.5. Characterisation of Chinook Salmon SOD1 Genes

Chinook salmon SOD1a (Appendix A; GenBank accession No. OP760294) and SOD1b (Appendix A; GenBank accession No. OP760295) cDNA coding sequences and genes were predicted using chromosomes LG33 and LG30, respectively. Both cDNA sequences were 465 nt in length and had a gene organisation of four exons and three introns, with the intron splicing consensus (GT/CAG) conserved at the 5′ and 3′ ends. Some synteny was found around the genomic regions where SOD1a and SOD1b was found when compared with the SOD1 region within other vertebrates (Appendix A). For salmonid SOD1a, TIAM Rac1 associated GEF 1 (TIAM1), SR-Related CTD associated factor 4 (SCAF4) and hormonally upregulated neu-associated Kinase (HUNK) were found either side of the SOD1 genes in most of the vertebrate genomes analysed. For salmonid SOD1b, HUNK and MIS18 Kinetochore Protein A (MIS18A) were found next to the SOD1 genes in most of the vertebrate genomes analysed. SOD1a and SOD1b of each salmonid were found on different chromosomes within the genomes, likely due to the salmonid-specific genome duplication event. Both Chinook salmon SOD1 cDNA’s encoded a 154 amino acid sequence, with no signal peptide predicted. Alignment of the predicted SOD1 protein sequences with selected known vertebrate SOD1 sequences, showed the presence of conserved regions, including two signature patterns for this family of enzymes; [GA]-[IMFAT]-H-[LIVF]-H-[S]-x-[GP]-[SDG]-x-[STAGDE] and G-[GNHD]-[SGA]-[GR]-x-R-x-[SGAWRV]-C-x(2)-[IV], two cysteine residues important in disulphide bond formation, a protein kinase C phosphorylation site and several amino acid residues required for the binding of copper and zinc (Figure 5). The close relationship of the Chinook salmon SOD1 with the other known fish SOD1 sequences, especially the rainbow trout and Atlantic salmon, is also apparent in the phylogenetic tree (Appendix A). Due to the high similarity of the Chinook salmon SOD1a and SOD1b sequences, primers were designed to amplify both expressed forms of the genes for qPCR (Appendix A).

### 3.6. Characterisation of Chinook Salmon HSP90 Genes

Chinook salmon HSP90AA1.1a (Appendix A; GenBank accession No. OP760297), HSP90AA1.2a (Appendix A; GenBank accession No. OP760296), HSP90AA1.1b (Appendix A; GenBank accession No. OP760298), HSP90AA1.2b (Appendix A; GenBank accession No. OQ215311), HSP90AB1a (Appendix A; GenBank accession No. OP760299) and HSP90AB1b (Appendix A; GenBank accession No. OP760300) cDNA coding sequences and genes were predicted using chromosomes LG05 and LG18. HSP90AA1.1a cDNA was 2208 nt in length, HSP90AA1.2a cDNA was 2181 nt in length, HSP90AA1.1b cDNA was 2202 nt in length and HSP90AA1.2b cDNA was 2178 nt in length, and all had a gene organisation of ten exons and nine introns. HSP90AB1a cDNA was 2181 nt in length and HSP90AB1b cDNA was 2172 nt in length and all had a gene organisation of eleven exons and ten introns. For all HSP90 genes, each intron contained the intron splicing consensus (GT/CAG) conserved at the 5′ and 3′ ends. A high degree of synteny around the Chinook salmon genomic regions where HSP90AA1.1a, HSP90AA1.2a, HSP90AA1.1b and HSP90AA1.2b were found when compared with the HSP90AA1 regions within other vertebrate genomes (Appendix A). Iodothyronine deiodinase 3 (DIO3), protein phosphatase 2, regulatory subunit b, gamma (PPP2R5C), WD repeat domain 20 (WRD20) and MOK protein kinase (MOK) were found either side of the HSP90AA1 genes in the majority of the vertebrate genomes analysed. Two copies (HSP90AA1.1a and HSP90AA1.1b) were found within the African clawed frog and Amazon molly genomes. Four copies were found within the rainbow trout and Chinook salmon genomes, with HSP90AA1.1a and HSP90AA1.2a tandemly arranged on one chromosome and HSP90AA1.1b and HSP90AA1.2b tandemly arranged on another. A high degree of synteny was also found around the Chinook salmon genomic regions where HSP90AB1a and HSP90AB1b were found when compared with the HSP90AB1 regions within other vertebrate genomes (Appendix A). NFKB inhibitor epsilon (NFKBIE), transmembrane protein 151B (TMEM151B) and T-complex associated testis expressed 1 (TCTE1) were found on one side of the HSP90AB1 genes in all the vertebrate genomes analysed. One copy of HSP90AB1 was found within the human, coelacanth, Amazon molly and elephant shark, whereas two copies were found within the rainbow trout and Chinook salmon genomes, with HSP90AB1a and HSP90AB1b of each salmonid found on different chromosomes within the genomes.

Chinook salmon HSP90AA1.1a cDNA sequence encoded a 735 amino acid sequence, HSP90AA1.2a cDNA a 726 amino acid sequence, HSP90AA1.1b cDNA a 733 amino acid sequence, HSP90AA1.2b cDNA a 725 amino acid sequence, HSP90AB1a cDNA a 726 amino acid sequence and HSP90AB1b cDNA a 723 amino acid sequence. None of these sequences were predicted to have a signal peptide; however, alignment of the predicted HSP90AA1 protein sequences with selected known vertebrate HSP90AA1 (Figure 6) and predicted HSP90AB1 protein sequences with selected known vertebrate HSP90AB1 sequences (Figure 7) showed the presence of conserved regions. These included a conserved heat shock hsp90 proteins’ family signature, Y-x-[NQHD]-[KHR]-[DE]-[IVA]-F-[LM]-R-[ED], an ATP-binding domain, containing a conserved “GxxGxG” motif at the N-terminal end and a MEEVD motif at the C-terminal end. The close relationship of the Chinook salmon HSP90AA1 and HSP90AB1 sequences with other known fish HSP90AA1 and HSP90AB1 sequences, especially rainbow trout, is apparent in the phylogenetic tree (Appendix A). Due to the complexity of designing primers to distinguish between the different Chinook salmon HSP90AA1 and HSP90AB1 sequences, primers were designed to amplify all expressed forms of the genes for qPCR (Appendix A).

### 3.7. Chinook Salmon SOD1 and HSP90 Expression during the Temperature/DO Trial

The expression of SOD1 and HSP90 was investigated in the liver spleen and gill tissues of fish that were exposed to the high-temperature/low-DO and normalised to either time-matched control or pre-trial control fish.

#### 3.7.1. SOD1

To show the effects of high-temperature and low-DO levels on the expression of SOD1 within Chinook salmon, the normalised fold expression of SOD1 was calculated using the time-matched control fish (Figure 8). Within the gill, expression of SOD1 was significantly (*p* < 0.05) upregulated in the high-temperature/low-DO fish (1.5-fold change) compared to the control fish. Within the liver, expression of SOD1 was significantly (*p* < 0.05) downregulated in the high-temperature/low-DO fish (1.9-fold change) compared to the control fish. Within the spleen, expression of SOD1 was significantly (*p* < 0.02) downregulated in the high-temperature/low-DO fish (5-fold change) compared to the control fish. Similar SOD1 expression was seen within Chinook salmon, when the normalised fold expression of SOD1 was calculated using the pre-trial control fish (Appendix A). The normalised fold expression of SOD1 was downregulated in the liver (1.6-fold change) and significantly (*p* < 0.02) downregulated in the spleen (3-fold change) due to increased temperature and low-DO levels. However, the gill expression of SOD1 was different, showing a significant (*p* < 0.02) decrease (3.7-fold change).

#### 3.7.2. HSP90

To show the effects of high-temperature and low-DO levels on the expression of HSP90 within Chinook salmon, the normalised fold expression of HSP90 was calculated using the time-matched control fish (Figure 9). Within the gill, expression of HSP90 was significantly (*p* < 0.05) downregulated in the high-temperature/low-DO fish (3.2-fold change) compared to the control fish. Within the liver, expression of HSP90 was significantly (*p* < 0.02) downregulated in the high-temperature/low-DO fish (7.8-fold change) compared to the control fish. Within the spleen, expression of HSP90 was significantly (*p* < 0.02) downregulated in the high-temperature/low-DO fish (7.9-fold change) compared to the control fish. Similar HSP90 expression was seen within Chinook salmon, when the normalised fold expression of HSP90 was calculated using the pre-trial control fish (Appendix A). The normalised fold expression of HSP90 was significantly (*p* < 0.05) downregulated in the gills (2.6-fold change), significantly (*p* < 0.02) downregulated in the liver (10.9-fold change) and significantly (*p* < 0.02) downregulated in the spleen (10.5-fold change) due to the high-temperature and low-DO levels.

## 4. Discussion

There is an existing body of research that has focused on temperature challenges in a range of fish species [2,48,59,86,87,88]. However, many of them focus on acute, short-term temperature changes, with experimental temperatures ranging between 5–16 °C higher than the controls [89] and an experimental duration normally shorter than 14 days [1,53,54,89,90,91,92,93]. Temperature changes due to global warming are gradual and likely to induce stress for prolonged periods of time, leading to chronic stress effects [94]. When developing biomarkers of stress, it is important to consider this, as there will be differences within the gene expression of fish that undergo short periods of stress (i.e., acute thermal stress) versus longer periods of exposure (i.e., chronic thermal stress) [1,95,96,97,98]. Additionally, it has been shown that chronic hypoxia has a significant impact on fish gene expression [99], with important differences being found in genes involved in lipid metabolism, immune responses, and steroid hormone release [100]. For these reasons, this investigation focused on a longer thermal challenge (up to 71 days) with a relatively small change in temperature (from 17 °C to a maximum of 20 °C) and lower DO. This was an attempt to mimic a warming environment where some Chinook salmon are farmed during summer periods.

### 4.1. General Health Parameters

Poikilotherms, such as Chinook salmon, are particularly vulnerable to thermal changes and their ability to cope with environmental variations and continuous stress is determined by their ability to elicit appropriate stress responses. Physiological changes, the re-allocation of energy towards defensive mechanisms and behavioural adaptations to cope with the threat, all play a role in this [2]. Physiological and behavioural processes change, leading to increased swimming activity and energy demand [101]. Additionally, biochemical reaction rates are regulated by temperature; a constant temperature increase will affect fish physiology at a biochemical level [95] and lead to the acceleration of the metabolic rate [102]. This has been demonstrated in *Sparus aurata*, where the expression of proteins related to the energetic pathway was increased in fish kept at 30 °C, suggesting a higher energy demand at higher temperatures [103]. In fact, fish require additional energy if the temperature exceeds their tolerance zone and remains high in order to overcome the adverse effects of temperature [2]. Examples of these effects, due to constant high temperature, include protein denaturation, DNA mutation, oxidative damage, and cellular death [104]. All these energy demands need to be overcome with a higher feeding rate. However, thermal stress is also known to reduce appetite, leading to the increased stressed state of the fish [23]. By entering this altered state, all non-critical functions (such as growth or reproduction) are slowed down to compensate for the most essential and immediate needs [2]. Therefore, thermal stress impacts physiological performance, in terms of the growth, size, and survival of the fish [104].

#### 4.1.1. Weight Change and Feed Intake

In this study, body weight, weight gain and HSI decreased significantly in fish exposed to high-temperature and low-DO compared to time-matched control fish; these results are consistent with the results of other studies [2,103,104,105]. Growth is an energetically demanding activity and is highly dependent on aerobic metabolism [106,107]. One possible explanation of fish weight loss could be the cost of adapting, through attempting to re-establish homeostasis and utilising energy for essential needs to survive, thus limiting the energy available for growth and non-survival needs. However, fish kept at the high-temperature and low-DO also had reduced feed intake compared to the time-matched control fish. Loss of appetite and reduced feeding activity in fish is a well-known result of stress [23]. In Atlantic salmon, high temperatures (above 20 °C) critically affected food consumption [108,109], with one study reaching a reduction of up to 67% [110]. In Rainbow trout facing a hypoxia challenge, the food intake was reduced by 50% [111].

#### 4.1.2. Mortality

There was a clear difference in fish mortality during the trial, with cumulative mortality reaching 22.0% by the end of the high-temperature and low-DO trial. By contrast, only two fish died in the time-matched control tanks, recording a cumulative mortality of just 4.6%. Higher mortality during the hottest summer months is well documented in New Zealand Chinook salmon [4,5,6,7,8,9], wild migrating Chinook salmon [112], trout [113,114], Sockeye salmon and Pink salmon [115] and a range of other species [116,117,118,119,120,121]. In some cases, an increase in the incidence of pathogens was also found during these periods, acting as an additional stressor to the fish. Chronic temperature challenges can act on fish: (1) as a stressor itself, (2) by altering the responses of the organism to further stressors or (3) by enhancing or generating new stressors, such as the increased presence of pathogens and disease [2]. However, this experiment was carried out in a RAS system supplied with disinfected seawater; the mortality was most likely due to the combined effects of high-temperature, low-DO and reduced feed intake, rather than as a result of infectious diseases.

### 4.2. Blood Health Parameters

A baseline of the blood haematology parameters found within healthy fish is essential for understanding the ranges considered as normal, or altered due to stress events. Studies on New Zealand Chinook salmon haematology and plasma biochemistry have been previously carried out and these tests have now been standardised [81]. This supports the relevance of these biomarkers in elucidating how Chinook salmon respond to stress events.

In fish, cortisol plays an important role in the stress response [122] and many studies have used cortisol as a biomarker to monitor the levels of stress in fish [123,124,125]. In this present study, cortisol levels in Chinook salmon kept at the high-temperature and low-DO were increased compared to the time-matched control and pre-trial fish; however, the differences were not statistically significant. This was not unexpected, as cortisol is the end product of the neuro–endocrine response to stress, which can make it an inaccurate biomarker, especially during chronic stress events. In addition, cortisol expression can be adversely affected by several other factors within a study, such as the genetics of an individual animal or the presence of unrecognised stressors, and so is not always reliable [126]. For example, the animals in this study were crowded prior to handling and euthanised, making the cortisol levels a possible result of the combined stress of handling and elevated temperatures, which has been previously reported [80]. Interestingly, in this investigation, fish kept at high-temperature and low-DO were found to have significantly higher lactate levels in their blood, compared to pre-trial and time-matched control fish. This result is consistent with the finding of Lulijwa et al. [80], who considered all the fish in this study. During anaerobic glycolysis, lactate is the end product after pyruvate reduction [127]. It has been shown, in trout undergoing increasing hypoxia, that there is a directly proportional increase in the levels of lactate relative to the DO level of the water [128]. Also, in juvenile Chinook salmon exposed to high temperatures (up to 24 °C), the plasma lactate concentrations were higher in treated fish compared to the controls [129]. This was associated with anaerobic metabolism, possibly correlated to decreased blood pH and the low affinity of haemoglobin and oxygen [129]. A similar result was reported in Atlantic salmon, where lactate was elevated at the peak of the thermal cycle and continued to be high for up to 8 h [47]. During our investigation, Chinook salmon not only experienced thermal changes, but were also exposed to different levels of DO, potentially contributing to the increased levels of lactate.

Immune parameters were also measured in the form of neutrophil, monocyte, and lymphocyte percentages. There was a significant increase in the percentage of neutrophils, a significant decrease in the percentage of lymphocytes and an increase in the ratio of neutrophils to lymphocytes within fish kept at high-temperature and low-DO compared to the pre-trial and time-matched control fish. Different levels of oxygen within the water have previously been reported to modulate lymphocyte levels within fish [130] and similar effects on neutrophil and lymphocyte percentages have been seen in other fish investigations [131]. In contrast, the haemoglobin and white blood cell count did not change in this study. We found no differences in the percentage of monocytes in the fish exposed to high-temperature and low-DO, and this has also been observed in other studies [132,133]. The increase in neutrophils indicates that an inflammatory response may be occurring in fish exposed to high-temperature and low-DO levels. A correlation between higher temperatures and inflammatory responses was found in *Sparus aurata* kept at 30 °C, where the fish had an ongoing inflammatory process, which was correlated with higher mortality at that temperature [103].

### 4.3. Gene Expression Study

A number of Chinook salmon-specific genes and their copies were identified from the available genome for use in qPCR. Before comparing the effects of high-temperature and low-DO on gene expression it was important to identify multiple gene copies present due to the fourth genome duplication that salmonids underwent [134,135]. Previous studies in fish with multiple copies of a gene have shown that each copy can have different expression patterns within the same tissue, and, more importantly, respond differently to external stimulants such as stress [136,137]. To increase the robustness of this study, SOD1 and HSP90 expression were normalised with two housekeeping genes, POLR2F and RPL18, with primers designed to detect all copies of these genes.

#### 4.3.1. Chinook Salmon POLR2F and RPL18

Two housekeeping genes, POLR2F and RPL18, were fully characterised. The POLR2F gene homologue encodes a subunit of RNA polymerase II, which is the polymerase responsible for synthesizing mRNA in eukaryotes [138], and is commonly used as a housekeeping gene candidate for qPCR [139]. Due to genome duplication events in salmonids, two POLR2F genes exist in Chinook salmon on different chromosomes. Good conservation of an acidic segment in the N-terminal region and a C-terminal region, essential for the enzymes’ activity [140] and the presence of the RNA polymerases K/14 to 18 Kd subunits’ signature [141], along with the synteny and phylogenetic analysis, clearly show these are POLR2F homologues. The RPL18 gene homologue encodes a ribosomal protein that belongs to the L18E family of ribosomal proteins and is a component of the ribosomes’ large 60S subunit [142] and is commonly used as a housekeeping gene candidate for qPCR [139]. Two RPL18 genes exist in Chinook salmon on different chromosomes due to genome duplication events in salmonids. Good conservation of a casein kinase II phosphorylation site, a N-myristoylation site, two amidation, a N-glycosylation site, cAMP-and cGMP-dependent protein kinase phosphorylation site, a protein kinase C phosphorylation site [143] and the presence of a ribosomal protein L18e signature [144], along with the synteny and phylogenetic analysis, clearly show these are RPL18 homologues. qPCR primers were designed to ensure amplification of all Chinook salmon POLR2F and RPL19 homologues in the diet trial.

#### 4.3.2. Chinook Salmon SOD1 and HSP90

Before SOD1 and HSP90 genes expression could be investigated, it was important to understand the number of copies Chinook salmon had of each gene before designing the appropriate primers.

##### SOD1

Two copies of SOD1 exist within the Chinook salmon genome, SOD1a and SOD1b, both of which contain an ORF of 465 nt, encoding a 154 amino acid sequence with no signal peptide predicted. Previous investigations have shown that SOD1 is present in other salmonids [145]; however, this is the first study to identify the existence of two copies of this gene in this group of fish. Good conservation of two signature patterns for this family of enzymes, two cysteine residues important in intramolecular disulphide bond formation, and several amino acid residues required for the binding of copper and zinc ions [146,147], along with the synteny and phylogenetic analysis clearly show these are SOD1 homologues. The SOD1a and SOD1b of each salmonid investigated were found on different chromosomes, likely due to the salmonid-specific genome duplication event [148].

The expression of SOD1 was found to be significantly downregulated within the liver and spleen of fish that had undergone the high-temperature and low-DO treatment when compared to the pre-trial and time-matched control fish. This has also been seen in the liver of Atlantic Salmon, where SOD1 expression was negatively affected in fish kept at a high temperature and low oxygen for 45 days [1]. Also, from the same study, a significantly lower transcriptional rate was seen in the individuals kept at high temperature compared to the fish kept at the control temperature, and showed metabolic depression, possibly saving energy to only transcribe vital proteins. In *Lateolabrax maculatus* kept at an increased temperature for a 96 h period, SOD1 expression was also decreased within the liver [149]. Oxidative stress in fish undergoing stress has also been demonstrated in several different species [56,57,150,151,152,153], including Chinook salmon [154]. In this study, SOD1 expression in the gills of fish kept at a high temperature and low DO differed when compared to the pre-trial control and time-matched control fish. When compared to the time-matched control fish, SOD1 expression was found to increase slightly, whereas it decreased when compared to the pre-trial fish. Because of this, it is difficult to draw any reliable conclusions with regards to the expression of SOD1 in the gills in this study and highlights some considerations to factor into experimental planning with regard to the controls and sampling of gills.

We found no other studies that have clearly looked at gene expression in the gills of different control populations of fish within an experiment; however, there are some observations to be made. Because the gills are an external organ with a large surface area in constant contact with the external environment, this makes them extremely sensitive to minor chemical or physical changes in their surroundings during experiments. In addition, due to their delicate nature, minor mechanical damage can occur, leading to downstream effects on experimental outcomes. A recent study has highlighted artefacts that can arise from sampling techniques that interfere with the interpretation of gill histology when studying them to understand diseases [155]. Similarly, in future studies measuring gill gene expression, ensuring that the control fish are not confounding gene expression profiles will also be important to consider. As seen in our study, this may not be critical for target organs, such as the spleen and liver, as the changes reflected there are more systemic and are not as sensitive to slight environmental changes.

##### HSP90

Four copies of HSP90AA1 and two copies of HSP90AB1 exist within the Chinook salmon genome. Good conservation of the HSP90 family signature [156], an ATP-binding domain containing a conserved “GxxGxG” motif, critical for binding ATP [157] and a MEEVD motif in the C-terminal, known to bind to tetratricopeptide repeat domain-containing proteins [158], along with the synteny and phylogenetic analysis, clearly show these are all HSP90AA1 and HSP90AB1 homologues. Interestingly, salmonids were not the only group of vertebrates to have multiple copies of HSP90AA1. The African clawed frog, coelacanth and the Amazon molly were all found to have two copies that appear to have come about due to a tandem gene duplication. In salmonids, four copies were found, with HSP90AA1.1a—HSP90AA1.2a and HSP90AA1.1b—HSP90AA1.2b both being positioned next to each other but on different chromosomes. The two copies that are adjacent to each other on the same chromosome are likely to be the result of a tandem duplication, as seen in Amazon molly [159], while the salmonid fourth whole genome duplication event [148] likely led to the two copies found on the two different chromosomes. The presence of HSP90AB1a and HSP90AB1b, is specific to salmonids and likely due to the salmonid-specific genome duplication event [148]. The presence of six HSP90 genes was known in Atlantic salmon and rainbow trout [49]; however, three copies had not been characterised in Chinook salmon [48].

In this investigation, HSP90 was downregulated within the liver, spleen, and gill of fish that had undergone the high-temperature and low-DO treatment, when compared to the pre-trial and time-matched control fish. This contrasts with previous studies that have reported HSP90 upregulation during heat stress [36,48,51,160,161]. In wild-caught Chinook salmon, a transcriptomic approach assessed HSP90 expression in fish kept at 18 °C and 20 °C degrees for 4 h and showed immediate activation of HSP90 following this acute stress [162]. A similar result was found in juvenile, ocean-type Chinook salmon kept for four days in multi-stressor conditions (with changes in temperature, DO and salinity), where HSP90 expression was identified as a potential thermal stress biomarker as it was upregulated at 18 °C [163]. However, these studies investigated acute thermal stress, where the timeframe of the experiments ranged from 1 h up to six days. In contrast, some studies have shown that HSP90 expression can decrease in the liver and the gills after an increase in temperature over a 96 h period at higher temperatures [149]. An investigation of thermal stress over seven days in Senegalese sole [164] showed downregulation of HSP90 at different time points of the experiment (1 h, 24 h, three days and seven days). Two copies of the Senegalese sole HSP90 were analysed, HSP90AA and HSP90AB, and their expression was examined in the liver, intestine, muscle, heart, gill, and brain. After seven days the expression of HSP90AA was downregulated in liver and muscle, whereas HSP90AB was downregulated in the liver and brain. The downregulation of HSP90AA was also seen in muscle (after 24 h), gills (after 1 h and after three days) and in the brain (after 1 h and after three days). For HSP90AB, downregulation was seen in the muscle (24h and three days), gills (1 h and three days) and brain (three days and seven days). Downregulation of HSP90 has also been reported in the eyed eggs of Chinook salmon reared until the fry stage at temperatures 3 °C higher than the control fish [165]. However, in two species of Pacific salmon (Sockeye and pink salmon), the expression of HSP90, after 5–7 days of constant high temperature (19 °C), was still upregulated [115]. This was explained by the fish not being able to acclimatise during the 5–7-day experiment. The different results in different species and life stages suggests that HSP90 expression in response to chronic thermal stress is species-, life-stage- and tissue-specific [166].

A possible explanation for the downregulation of HSP90 in Chinook salmon could be the acclimatisation of the fish to the elevated temperature and lower DO. This has been previously reported in *S. aurata* kept at 24 °C for one month, where a study of the proteome suggested a low level of cellular stress, suggesting a non-stressful status of the fish [103] despite the 15% mortality observed. In this present study, the mortality, general health parameters, lactate elevation and weight loss all suggest that adult Chinook salmon kept at 19–20 °C for 71 days were stressed, making acclimatisation an unlikely explanation for the downregulation of HSP90. Another possible reason could be related to the chronically stressed status of the fish and channelling of energy into essential functions [2]. It is possible that during chronic stress the fish developed pre-transcriptional mechanisms to prevent the translation of non-essential genes into mRNA, to save energy. This concept has previously been demonstrated in Chinook salmon exposed to stress, where the reduced mRNA level of growth biomarkers (IGFI, IGFBP2b and THR-B) and mobilisation of energy and gluconeogenesis biomarkers (FAS and PEPCK) showed a reduction in energy allocated to growth [167]. Therefore, channelling energy for vital activities during prolonged stress, could explain the downregulation of HSP90 in Chinook salmon after 71 days at high temperatures and lower DO. Confirming this hypothesis, additional sample analysis showed that very low levels of total protein, glucose, cholesterol, creatine kinase (CK), acid phosphate (ACP) and alkaline phosphate (ALP) were found in the fish that lost weight during this study [80]. These changes have been correlated in other studies with long-term starvation, malnutrition, and impaired energy metabolism, as well as muscle weakening and reduced ATP turnover [80].

To better understand the physiological changes of Chinook salmon during a long-term temperature and oxygen challenge, it would be beneficial to include some additional assessments and sampling throughout the trial. For example, having additional sampling within this trial at 7 and 35 days prior to the end of the trial would have provided a better understanding of how HSP90 expression was changing over time and during the initial acclimation. However, this would need to be balanced against the additional handling stress and possible elevated mortality this could cause. Additionally, there are a range of other markers that could be targeted that have been identified in other salmonid studies looking at heat stress and hypoxia, such as HSP70 [3,48,168,169]. However, as was done in this investigation with the HSP90 family in Chinook Salmon, a full characterisation of all gene copies would be a prerequisite before designing primers for expression work.

## 5. Conclusions

The results of this long-term temperature and DO challenge demonstrated that the physiology of Chinook salmon is compromised by temperatures 2–3 °C above their optimal range in combination with lower DO, a response also seen in other fish species [2]. The challenge also provided insights into how these fish will be affected by increasing temperatures and lower DO levels due to global warming. Climate change is a major threat for many marine organisms and understanding how farmed fish will cope with the effects is essential in order to develop effective mitigation and adaptation plans. To elucidate the mechanisms underlying the effects of heat stress, it is important to consider molecular approaches alongside the observed physiological changes, as demonstrated in this present study. Our results showed that plasma cortisol levels were not an unequivocal biomarker of chronic thermal stress, but plasma lactate levels were a more reliable candidate. Increased HSP90 expression has been widely utilised as a biomarker of acute thermal stress, but its variable response and observed downregulation during chronic thermal stress means the results need to be carefully analysed and contextualised appropriately within future experiments. SOD1 expression was downregulated during the temperature and DO challenge and could be utilised as an oxidative stress and thermal stress biomarker, but more targeted experiments need to be undertaken to understand how SOD1 responds specifically to hypoxia or temperature in separate Chinook salmon challenges.

Predicting and understanding how Chinook salmon react to environmental changes will help farmers make informed decisions on appropriate future farm locations. The proposed move to farm Chinook salmon in open ocean pens in New Zealand [170], could help to resolve the effects that climate change would have on this important industry.

## Figures and Tables

**Figure 1 biology-12-01342-f001:**
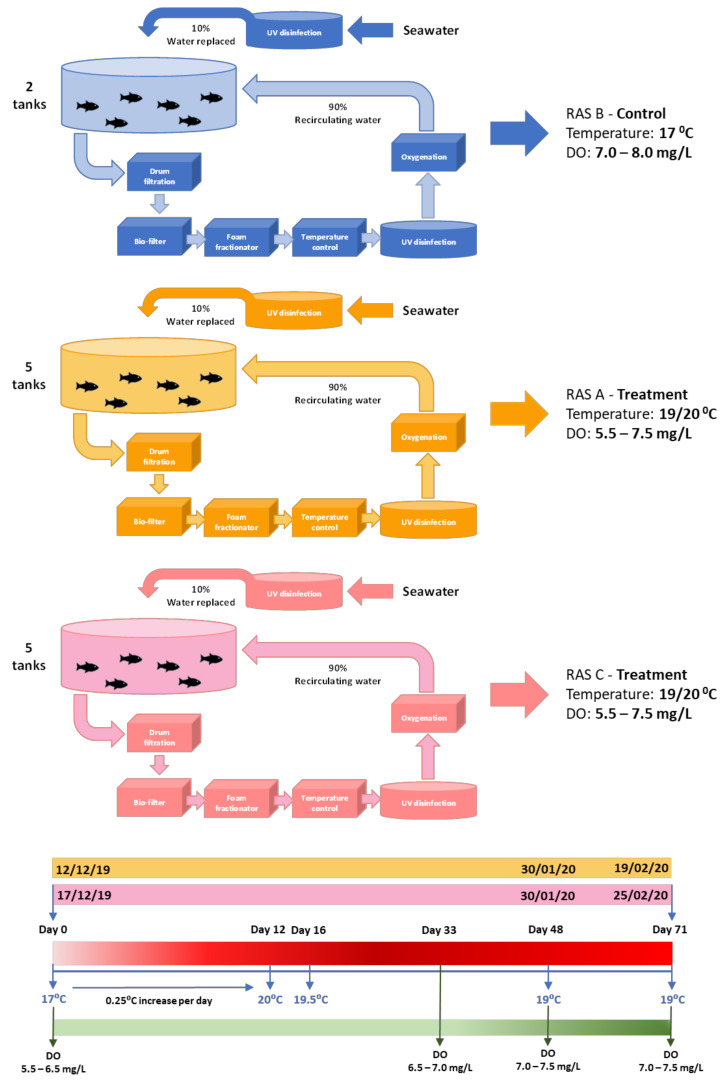
RAS system experimental set up, showing the timeline of the main events that occurred during the trial.

**Figure 2 biology-12-01342-f002:**
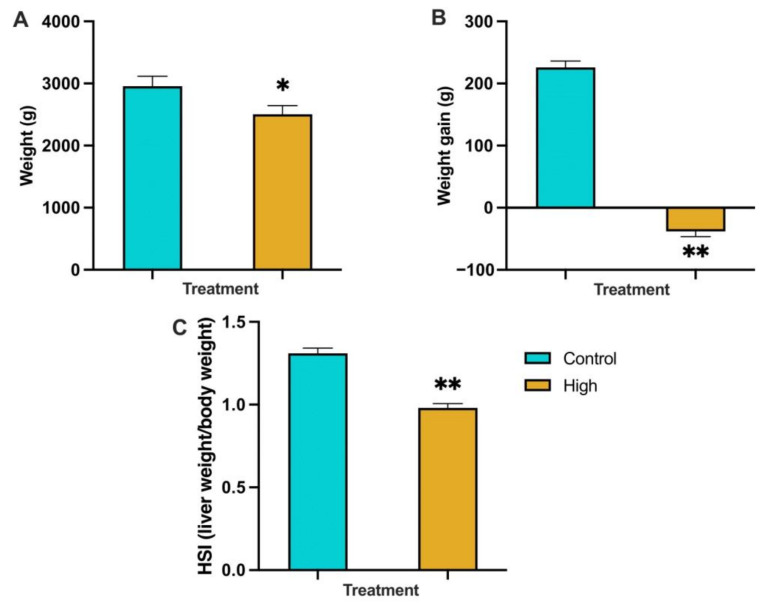
The mean weight (**A**); mean weight gain/loss (**B**); and HSI (**C**) of Chinook salmon at the end of the trial, sampled from the control-temperature (Control) tanks (*n* = 16) or the high-temperature and low-DO (High) tanks (*n* = 16). ** indicates *p* < 0.02. * indicates *p* < 0.05. Error bars represent the standard error.

**Figure 3 biology-12-01342-f003:**
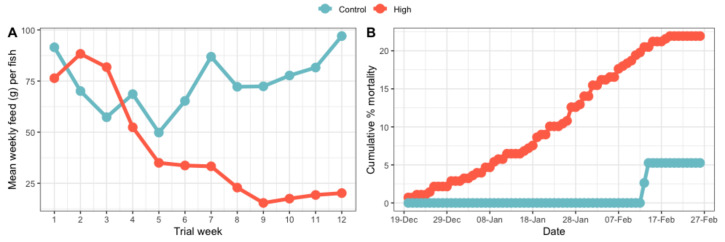
The weekly tank feed intake (**A**); and percentage cumulative mortality (**B**) of fish within the high-temperature and low-DO (High) tanks or the control-temperature (Control) tanks during the trial. Feed intake was measured daily after each meal and subsequently analysed as the mean daily feed intake (g) per fish per week. Results for RAS system B (control) are the mean of the two tanks. Results for the high-temperature and low-DO are the mean of 10 tanks from RAS A and C.

**Figure 4 biology-12-01342-f004:**
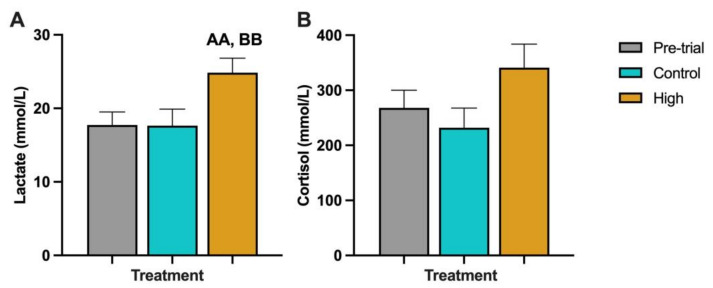
The lactate (**A**); and cortisol (**B**) levels found in the blood of Chinook salmon taken from the pre-trial (Pre-trial) tanks (*n* = 16), time-matched control temperature (Control) tanks (*n* = 16) or the high-temperature and low-DO (High) tanks (*n* = 16). AA indicates *p* < 0.02 between high-temperature/low-DO and pre-trial. BB indicates *p* < 0.02 between high-temperature/low-DO and time-matched control temperature.

**Figure 5 biology-12-01342-f005:**
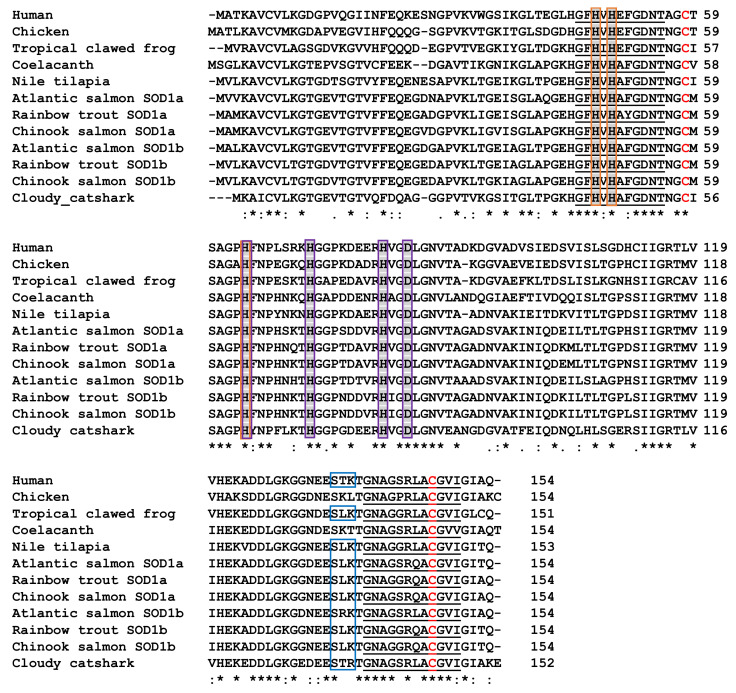
Multiple alignment of the predicted Chinook salmon SOD1a and SOD1b amino acid sequences with known vertebrate SOD1 molecules. Identical (*) and similar (: or .) residues identified by the Clustal omega program are indicated. The two Cu/Zn SOD family signatures are underlined (_) and the protein kinase C phosphorylation site is boxed in blue. The amino acids required for binding of copper (His-47, -49, -64, and -121) are shaded and boxed in orange and zinc (His-64, -72, and -82 and Asp-84) are shaded and boxed in purple. Two cysteines (Cys-58 and Cys-147) predicted to be engaged in disulphide bond formation are highlighted in red. GenBank accession no. Human, P00441.2; Chicken, P80566.3; Tropical clawed frog, AAI21541.1; Coelacanth, XP_006002003.1; Nile tilapia, XP_003446855.1; Rainbow Trout SOD1a, NP_001117801.1; Rainbow Trout SOD1b, NP_001154086.1; Atlantic Salmon SOD1a, NP_001117059.1; Atlantic_Salmon_SOD1b, XP_014053858.1; Chinook salmon SOD1a, OP760294; Chinook salmon SOD1b, OP760295; Cloudy catshark, ABJ53250.1.

**Figure 6 biology-12-01342-f006:**
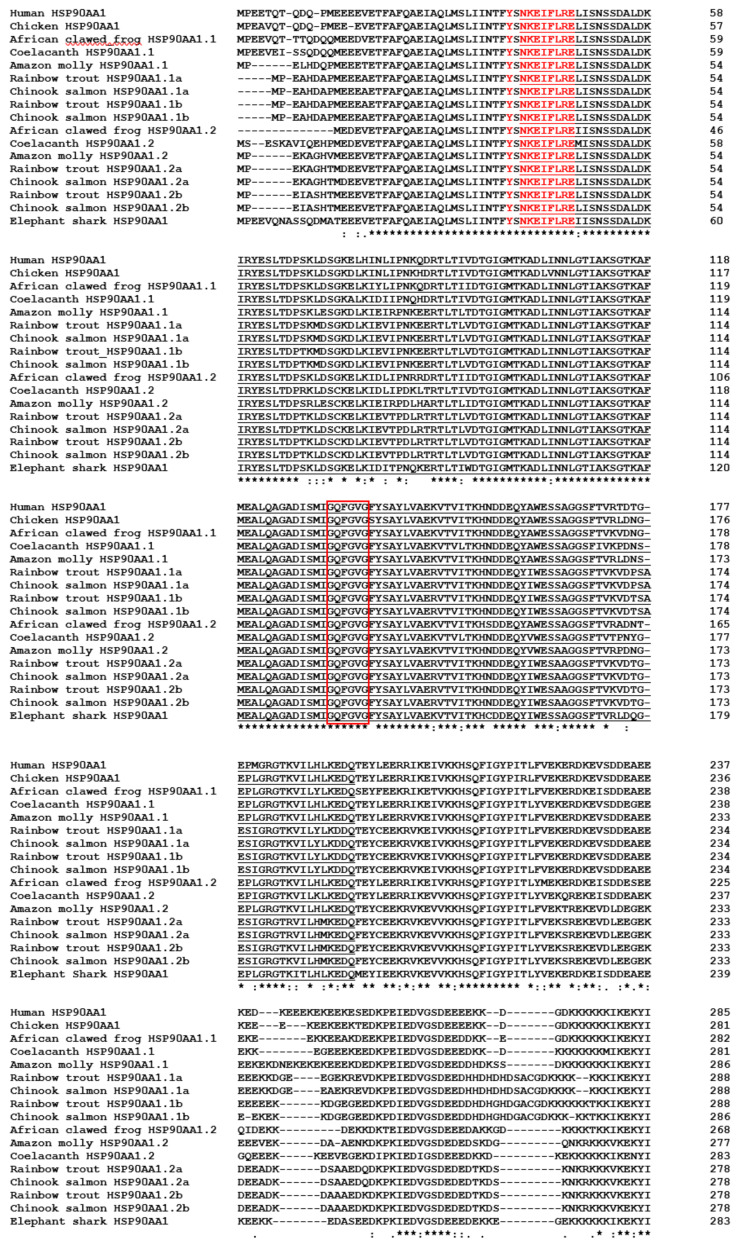
Multiple alignment of the predicted Chinook salmon HSP90AA1.1 and HSP90AA1.2 translations with selected vertebrate HSP90AA1.1 and HSP90AA1.2 molecules. Identical (*) and similar (: or .) residues identified by the Clustal omega are indicated. The ATP binding domain is underlined. The heat shock hsp90 proteins’ family signature is highlighted in red, the conserved “GxxGxG” motif is boxed in red and the MEEVD consensus sequence is highlighted in grey and boxed in blue. The accession numbers of the genes are: Human HSP90AA1, NP_001017963.2; Chicken HSP90AA1, NP_001103255.1; African clawed frog HSP90AA1.1, XP_031746306.1; African clawed frog HSP90AA1.2, NP_001072765.1; Coelacanth HSP90AA1.1, XP_014347228.1; Coelacanth HSP90AA1.2, XP_006010474.1; Amazon molly HSP90AA1.1, XP_007577309.1; Amazon molly HSP90AA1.2, XP_007577310.1; Rainbow trout HSP90AA1.1a, XP_021468114.1; Chinook salmon HSP90AA1.1a, OP760297; Rainbow trout HSP90AA1.1b, Chinook salmon HSP90AA1.1b, OP760298; Rainbow trout HSP90AA1.2a, CDQ81679.1; Chinook salmon HSP90AA1.2a, OP760296; XP_021468115.2; Rainbow trout HSP90AA1.2b, XP_021456592.1; Chinook salmon HSP90AA1.2b, OQ215311; Elephant shark HSP90AA1, XP_007886533.1.

**Figure 7 biology-12-01342-f007:**
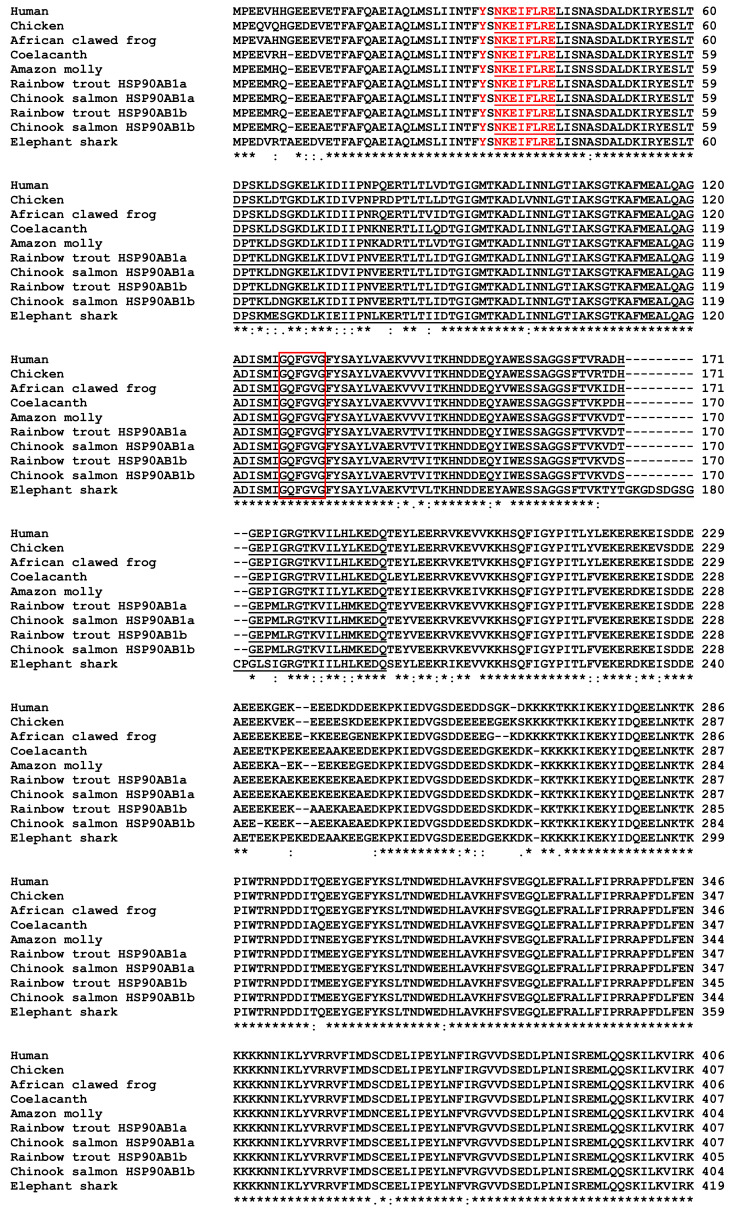
Multiple alignment of the predicted Chinook salmon HSP90AB1a and HSPAB1b translation with selected vertebrate HSP90AB1 molecules. Identical (*) and similar (: or .) residues identified by Clustal omega are indicated. The ATP-binding domain is underlined. The heat shock hsp90 proteins’ family signature is highlighted in red, the conserved “GxxGxG” motif is boxed in red and the MEEVD consensus sequence is highlighted in grey and boxed in blue. The accession numbers of the HSP90AB1 proteins are: Human, P08238.4; Chicken, NP_996842.1; African clawed frog, XP_041418547.1; Coelacanth, XP_014350000.1; Amazon molly, XP_007545616.1; Rainbow trout HSP90AB1a, CDQ78137.1; Rainbow trout HSP90AB1b, NP_001118063.1; Chinook salmon HSP90AB1a, OP760299; Chinook salmon HSP90AB1b, OP760300; Elephant shark, XP_007896400.1.

**Figure 8 biology-12-01342-f008:**
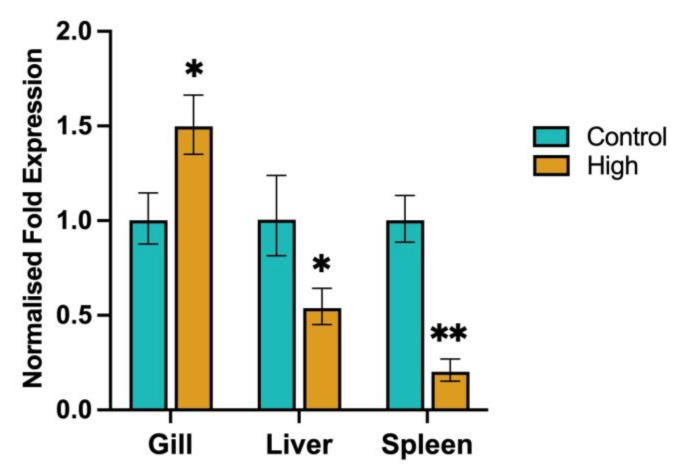
Normalised fold expression of SOD1 in the liver, spleen, and gill of time-matched control fish (Control) and fish kept at high-temperature/low-DO (High). Error bars represent the standard error. ** indicates *p* < 0.02 and * indicates *p* < 0.05.

**Figure 9 biology-12-01342-f009:**
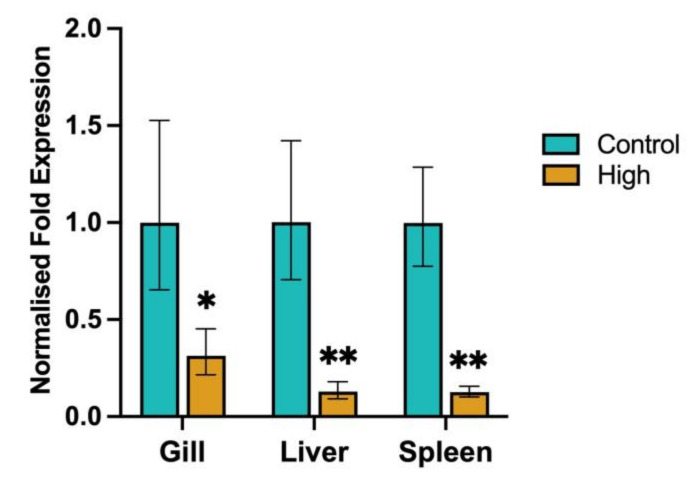
Normalised fold expression of HSP90 in the liver, spleen, and gill of time-matched control (Control) fish and fish kept at the high-temperature/low-DO (High). Error bars represent the standard error. ** indicates *p* < 0.02 and * indicates *p* < 0.05.

**Table 1 biology-12-01342-t001:** Genomes from different species used to analyse the level of gene synteny.

Species	Genome Version	Accession Number
Human	GRCh38.p14	GCA_000001405.29
Chicken	GRCg6a	GCA_000002315.5
Turkey	Turkey_5.1	GCA_000146605.4
Chinese softshelled turtle	PelSin_1.0	GCA_000230535.1
Green anole	AnoCar2.0	GCA_000090745.2
*Xenopus laevis*	Xenopus_laevis_v10.1	GCA_017654675.1
*Xenopus tropicalis*	UCB_Xtro_10.0	GCA_000004195.4
Coelacanth	LatCha1	GCA_000225785.1
Amazon molly	Poecilia_formosa-5.1.2	GCA_000485575.1
Rainbow trout	Omyk_2.0	GCA_025558465.1
Atlantic Salmon	Ssal_v3.1	GCA_905237065.2
Nile tilapia	O_niloticus_UMD_NMBU	GCA_001858045.3
Elephant shark	Callorhinchus_milii-6.1.3	GCA_000165045.2

**Table 2 biology-12-01342-t002:** Chinook salmon SOD1, HSP90, POLR2F, and RPL18 specific primers designed for qPCR.

Gene	Primer Name	Sequence (5′-3′)	Product Size	Annealing Temperature
SOD1	SOD1-F	GAGCTGACAATGTGGCTAAGA	101 bp	60 °C
	SOD1-R	CAGCTTTCTCATGGATCACC		
HSP90	HSP90-F	CTTTGAGAACAAGAAGAAGAAGAAC	92 bp	65 °C
	HSP90-R	CACACCCTTAATGAAGTTGAGGTAC		
POLR2F	POLR2F-F	AGAAGAGGATCACAACCCAATAC	162 bp	61 °C
	POLR2F-R	GGGATCTTCCTGCACTTTAGC		
RPL18	RPL18-F	GCTCTGAAGGTGACTGACGG	206 bp	61 °C
	RPL18-R	TTGGAACGAATGTAGGGCTTGG		

**Table 3 biology-12-01342-t003:** Various blood parameters measured, showing the mean values and standard error between fish in the pre-trial control (Pre-Trial), time-matched control (Control) and high-temperature and low-DO (High-Temp) tanks. ^aa^ indicates *p* < 0.02 between High-Temp and Pre-Trial. ^bb^ indicates *p* < 0.02 and ^b^ indicates *p* < 0.05 between High-Temp and Control.

Blood Parameter	Pre-Trial (*n* = 16)	Control (*n* = 16)	High-Temp (*n* = 16)
Blood Haemoglobin (g/L)	106.81 ± 3.41	101.18 ± 3.42	102.12 ± 3.44
White blood cell count (109/L)	25.00 ± 3.80	27.91 ± 3.78	25.78 ± 3.81
Monocytes (%)	1.25 ± 0.22	0.93 ± 0.16	0.87 ± 0.21
Lymphocytes (%)	92.19 ± 1.55	91.56 ± 1.32	81.94 ± 1.36 ^aa, b^
Neutrophils (%)	6.56 ± 0.81	7.50 ± 0.87	17.19 ± 1.10 ^aa, bb^
Neutrophil/Lymphocyte ratio	0.06 ± 0.01	0.08 ± 0.01	0.26 ± 0.04 ^aa, bb^
Mean corpuscular haemoglobin (%)	25.88 ± 0.73	28.70 ± 0.77	26.28 ± 0.75

## Data Availability

The sequence data that support the findings of this study are openly available in GenBank at NCBI (https://www.ncbi.nlm.nih.gov), under accession no. OQ215307, OQ215308, OQ215309, OQ215310, OP760294, OP760295, OP760296, OP760297, OP760298, OP760299, OP760300, OQ215311.

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
