# Peer review of "Characterising the Physiological Responses of Chinook Salmon (*Oncorhynchus tshawytscha*) Subjected to Heat and Oxygen Stress"

_biology, 2023, doi:10.3390/biology12101342_

Round 1

Reviewer 1 Report

The MS entitled “Characterising the physiological responses of Chinook salmon (Oncorhynchus tshawytscha) subjected to heat and oxygen stress” is an interesting study about the physiology of Chinook salmon. The MS is well planned, organised and studied few gene expression against the stress.  The abstract, introduction, methodology and result section are well written. The data analysis is in a good manner and the discussion part is very informative. The MS is acceptable for publication

Author Response

We appreciate the time and effort it took the reviewer to go through the manuscript and appreciate the overall positive comments. Thankyou.

Reviewer 2 Report

The manuscript entitled: “Characterising the physiological responses of Chinook salmon (Oncorhynchus tsawytscha) subjected to heat and oxygen stress” deals with a study of the response of cultured salmon in chronic conditions of high temperature and low oxygen. The analyzed parameters have been haematological, physiological, and of genetic markers relative to the expression of critical genes such as HSP90 and SOD1 (and related POLR2F/RPL18).

 In principle the research is not new in the references panorama, even if the authors have ignored some key papers and also ignored the study of HSP70 expression (transcription and translation and action of gene); however, the species studied is new and could be interesting to know the response of this oceanic fish species.

In particular, I invite the authors to follow these suggestions:

1)Introduction

a) Lines 51-70: the authors should summarize this part in a maximum of 10 lines.

b) The references lack some important manuscripts That should be read and cited in the introduction and the discussion (Animals. 2023; 13(8):1340. https://doi.org/10.3390/ani13081340¸Marine and Freshwater Behaviour and Physiology 43(4):283-296-DOI: 10.1080/10236244.2010.504046; Gen Comp Endocrinol. 2001 Oct;124(1):97-105. doi: 10.1006/gcen.2001.7688. PMID: 11703075; Fishes 2017, 2, 16. https://doi.org/10.3390/fishes2030016; Fish Physiol Biochem. 2019 Jun;45(3):997-1013. doi: 10.1007/s10695-019-00614-9. Epub 2019 Feb 4. PMID: 30715663.

c) Lines 85-94 HSPs are relative to the stressors for the cells. I can agree on the utilization of HSP90. Still, it is necessary to use also the HSP70 marker because the manuscript: has clearly defined the translation of HSP70 is enhanced in oxygen and temperature changes (J clinical Invest 2005(10) 2633-2639- DOI: 10.1172/JCI26471). Thus, the analysis should also be considered this marker, in view to also analyze the apoptosis and immune deficiency induced by heat/hypoxia stressors (Fishes 2017, 2, 16. https://doi.org/10.3390/fishes2030016).

d) Line 265.  The MDPI journal undergoes the EU rules. AQUI-S (New Zealand Ltd, NZ) is not accepted in the protocols allowed to euthanize the fish (https://en.wikipedia.org/wiki/Directive_2010/63/EU; see:  22 September 2010 the EU adopted Directive 2010/63/EU). This can be a problem and it is the task of the Editor to accept this euthanasia system. Thus, I ask the authors to give in the materials and methods the necessary information on the chemical and the New Zealand agreement for this research.

Results

a)           The results are too long and widely written. The authors probably could shift some pictures and data in the supplementary file (like the HSP90 isoform-lineaments with other species).

b)           Table 3: monocytes decrease in blood because they changed in macrophages in tissues due to an enhancement of apoptosis. The authors should discuss this data.

Discussion

a)           It is too long and widely written.

b)           The downregulation of HSP90 could be related to an enhancement of HSP70, but the data are not presented.

Author Response

Below are the responses to this reviewer:

1. Introduction

 a) Lines 51-70: the authors should summarise this part in a maximum of 10 lines.

The reviewer is asking for the following to be summarised in 10 lines:

Within global marine environments where temperatures are increasing, the com-bination of thermal stress and potential hypoxia have begun to have drastic effects on fish health and aquaculture production, creating an environmental challenge for the aquaculture industry [1-3]. It is widely accepted that ocean temperatures have increased over the last few decades and this is predicted to continue [4]. This has been especially noticeable during summer periods in many countries, including New Zea-land, where record temperatures are being reached [5]. Although the impacts depend on location and season, around New Zealand and in the Tasman Sea, there is a long-term trend towards warmer oceans [6]. The optimal temperature for New Zealand Chinook salmon (Oncorhynchus tshawytscha) is between 15-17 °C, where the best health and growth of the animals are achieved [7]. However, within the Marlborough Sounds, where many Chinook salmon are farmed, the last five years have seen peak summer-time water temperatures exceed 18°C [8, 9] at some farm sites and increased mortali-ties have been reported. Higher mortality correlated with heat waves and chronic thermal stress is not unexpected as this has been observed in other salmonid species, such as rainbow trout, where high temperatures are considered the main reason for a mortality syndrome during summer [10]. Fish can adapt and cope with daily and sea-sonal water temperature variations [11], but due to accelerated global warming, farmed poikilothermic fish, such as salmonids, could be under threat.

We believe this section justifies the research to the reader, and provides the overall context as to why the study was being undertaken. It would of helped if the reviewer had provided a reasoning behind why it should be reduced, so we could understand what they did not like about it. We would like to leave it as it is, but are happy to change it if the editor has some insight into what specifically should be omitted.

b) The references lack some important manuscripts that should be read and cited in the introduction and the discussion

We have listed below the references suggested by the reviewer and have gone through each one to determine their relevance to this particular study:

i) Athanasios Samaras (2023) A Systematic Review and Meta-Analysis of Basal and Post-Stress Circulating Cortisol Concentration in an Important Marine Aquaculture Fish Species, European Sea Bass, Dicentrarchus labrax. Animals. 2023; 13(8):1340. doi.org/10.3390/ani13081340¸

This is a very specific meta-analysis of data collected within Seabass, which is completely unrelated to a salmonid and more specifically Chinook Salmon, that was the focus of our study. Also the aim of this meta-analysis was to look at the variability of cortisol results between studies, which was found to be high. We are unable to see the relevance for this study for our investigation. However, there is a good review mentioned in this study that lists possible reasons why cortisol levels should be used with caution and we have included this into our discussion:

“Ellis, T.; Yildiz, H.Y.; López-Olmeda, J.; Spedicato, M.T.; Tort, L.; Øverli, Ø.; Martins, C.I.M. Cortisol and finfish welfare. Fish Physiol. Biochem. 2012, 38, 163–188.”

ii) Carbonara et al., (2010) The effects of stress induced by cortisol administration on the repeatability of swimming performance tests in the European sea bass (Dicentrarchus labrax L.) Marine and Freshwater Behaviour and Physiology 43(4):283-296-DOI: 10.1080/10236244.2010.504046;

This study is to do with the administration of cortisol to fish and looking at swimming performance in seabass. We can see no relevance of this investigation to the work undertaken in our study.

iii) Basu et al., (2001) The effects of cortisol on heat shock protein 70 levels in two fish species. Gen Comp Endocrinol. 124(1):97-105. doi: 10.1006/gcen.2001.7688.

We agree with the reviewer that HSP70 is another candidate that has potential as a good biomarker for stress within fish. However, it was not included in this investigation which is addressed under comment (c).

iv) Romano et al., (2017) Water Oxygen Content Affects Distribution of T and B Lymphocytes in Lymphoid Tissues of Farmed Sea Bass (Dicentrarchus Labrax). Fishes, 2, 16. doi.org/10.3390/fishes2030016

Again, this study is carried out in Seabass which is not closely related to a salmonid, but does report that oxygen levels can affect lymphocyte levels in fish, so we have included it in the discussion.

v) Abdel-Tawwa et al., (2019) Fish response to hypoxia stress: growth, physiological, and immunological biomarkers. Fish Physiology and Biochemistry volume 45, pages997–1013 doi: 10.1007/s10695-019-00614-9.

This review has already been referenced within the introduction of this investigation, to highlight the effects of hypoxia on lactate levels in fish.

c) Lines 85-94 HSPs are relative to the stressors for the cells. I can agree on the utilization of HSP90. Still, it is necessary to use also the HSP70 marker because the manuscript: has clearly defined the translation of HSP70 is enhanced in oxygen and temperature changes (Beere (2005) Death versus survival: functional interaction between the apoptotic and stress-inducible heat shock protein pathways. J clinical Invest 115:2633-2639. Thus, the analysis should also be considered this marker, in view to also analyze the apoptosis and immune deficiency induced by heat/hypoxia stressors (Fishes 2017, 2, 16. https://doi.org/10.3390/fishes2030016).

Whilst we agree that HSP70 would be another good marker to consider for use as a biomarker of stress in fish, this investigation already contains a significant amount of work and has shown the importance of properly characterising all possible copies of genes in a salmonid, before designing primers for expression work. In mammals, 11 different HSP70 family member genes have been characterised, so a significant amount of work would be required to first characterise all members of the HSP70 family in a salmonid, such as Chinook salmon. Once this is done, it would then need to be decided which HSP70 member to target and develop the appropriate primers. Because this has not yet been done in any species, I do question whether all previous studies carried out that say they have looked at HSP70 expression, truly understand which member of the HSP70 family they have looked at or can state the primers they have used are specifically looking at the expression of one or more genes. This is by no means a criticism of previous research, but something we now have more insight about, given the availability of genomes. However, to address the reviewers concerns, we have made a statement in the discussion near the end that HSP70 could be another good marker to consider for future studies. But similar to what we have done with HSP90, a full characterisation of this family of genes would be a prerequisite, before designing primers for expression work. We have included references to support this.

d) Line 265. The MDPI journal undergoes the EU rules. AQUI-S (New Zealand Ltd, NZ) is not accepted in the protocols allowed to euthanize the fish (https://en.wikipedia.org/wiki/Directive_2010/63/EU; see: 22 September 2010 the EU adopted Directive 2010/63/EU). This can be a problem and it is the task of the Editor to accept this euthanasia system. Thus, I ask the authors to give in the materials and methods the necessary information on the chemical and the New Zealand agreement for this research.

In response to the reviewers concerns with regards the method used to euthanize fish, the following should help address this. AQUI-S® (New Zealand, LTD) is approved for use in fish in Australia, New Zealand, Chile, Norway, Iceland and the Faroe Islands (Aqui-S website), and its active agent, isoeugenol, was approved in the European Union in 2011 (European Commission, 2011, regulation No 363/2011). This anaesthetic has been tested in many aquaculture species (Javahery and Moradlu, 2012; Soldatov 2021).

Javaheri, S. and Moradlu, A.H., AQUI-S, a new anesthetic for use in fish propagation, Global Vet., 2012, vol. 9, no. 2, p. 205.

Soldatov, A.A. Functional Effects of the Use of Anesthetics on Teleostean Fishes (Review). Inland Water Biol 14, 67–77 (2021). https://doi.org/10.1134/S1995082920060139

The authors would be happy to include this within the manuscript if the editor feels it is necessary to justify the use of AQUI-S® (New Zealand, LTD) in this study.

2. Results

a) The results are too long and widely written. The authors probably could shift some pictures and data in the supplementary file (like the HSP90 isoform-lineaments with other species).

When putting together the manuscript, the authors seriously considered the relevance of what was included within the main manuscript. As can be seen, there is already a significant amount of the figures for the results in the supplementary file. We have only included what we feel highlights to the reader the most important results. The alignments are useful, because it shows all the genes present for HSP90 and SOD1 within Chinook Salmon and clearly shows their close relationship to one another.

With regards to what has been written, there are a lot of results presented, both in the manuscript and within the Supplementary file which I can see, makes it feel as if there has been too much written. Given the other two referees did not raise specific issues with the results and how they were presented, we would prefer to leave them as they are, unless the editor has some specific suggestions around this matter.

b) Table 3: monocytes decrease in blood because they changed in macrophages in tissues due to an enhancement of apoptosis. The authors should discuss this data.

This was slightly confusing, because our results showed there was no change in monocyte numbers in this study, which we highlight in the discussion and mention other studies that have seen similar things. Regardless of this, it would have been highly speculative to have talked about them changing into tissue macrophages, due to apoptosis as we had not measured tissue macrophage numbers or looked at apoptosis.

3. Discussion

a) It is too long and widely written.

Again, similar to the statement about the results, this is a very generalised statement and difficult to address, if there is no clear part that the reviewer feels needs to be addressed. The authors feel that the discussion attempts to look at the results in the light of other studies and provides the reader with the relevant information that covers the findings in this investigation. Given the other two referees did not raise specific issues with the discussion, we would prefer to leave them as they are, unless the editor has some specific suggestions around this matter.

b) The downregulation of HSP90 could be related to an enhancement of HSP70, but the data are not presented.

The authors agree that the downregulation of HSP90 could be an enhancement of HSP70, but equally there could also be a downregulation of HSP70, the truth is we don’t know, especially when Chinook Salmon are under chronic stress. We don’t believe it would enhance the investigation to start speculating on the behaviour of other genes, however we do agree with the reviewer that HSP70 would be an interesting candidate in the future to also characterise and include in studies such as this and have addressed this in the discussion, as mentioned above.

Reviewer 3 Report

The paper provides a detailed examination of the Chinook salmon's response to sustained thermal stress and decreased dissolved oxygen levels, effectively employing both physiological assessments and gene expression studies. The structure of the paper reads great - from gene identification and phylogenetic analysis to the implementation of a experiments and data interpretation, coupled with the extensive supplementary materials and schematic figures.

Despite the paper's overall excellence, I have a couple of minor suggestions.

1. Adding a concluding sentence in the abstract to emphasize the significance of the study. Just for an example, something like "Our findings enrich the understanding of Chinook salmon's physiological and genetic responses to heat stress, key for sustainable fish farming in an era of changing global climates." (please use your own sentence).

2. line 159: For clarity, insert a link to the tree in the supplementary.

3. Figures 2 and 4: the confidence intervals/standard error look strange - please correct them.

Author Response

Below are our responses to the reviewers comments:

1. Adding a concluding sentence in the abstract to emphasize the significance of the study. Just for an example, something like "Our findings enrich the understanding of Chinook salmon's physiological and genetic responses to heat stress, key for sustainable fish farming in an era of changing global climates." (please use your own sentence).

We appreciate the reviewers suggestion and have added the statement “This study provides important insights into the physiological and genetic responses of Chinook salmon to temperature and oxygen stress, which are going to be critical for developing sustainable fish aquaculture, in an era of changing global climates.”

2. Line 159: For clarity, insert a link to the tree in the supplementary.

This refers to the methodology used for drawing the phylogenetic trees, so it is not usual to reference the results obtained in the methods. We have clearly referenced each tree found in the supplementary file when we present it within the results.

3. Figures 2 and 4: the confidence intervals/standard error look strange - please correct them.

     Figures 2 and 4 have been updated and the standard errors corrected. These have been added to the manuscript and the original files provided.

Round 2

Reviewer 2 Report

The Authors have partially improved the manuscript. However, the results and discussion sections should be shortened (more than 5 Discussion pages).

The English is fine, please look for long sentences.

Author Response

We really appreciate the the time and effort the reviewer has taken to look through the manuscript and help strengthen our work. However, as we stated before, there are a significant amount of results in this paper, as we have characterised not only the HSP90 genes, but also the housekeeping genes that exist in Chinook salmon. This was to ensure that the primers designed took into account all copies of genes that were present when undertaking the realtime results. On top of that we also have the physiological parameters that were measured and were included. We believe this is why the results and discussion appear larger than is normally seen in a manuscript, especially because there is a lot of results in the 30 suplementary figures we have, which we talk about. In addition, the discussion size was increased futher so we could include important points that this reviewer raised.

What would really help us would be some direction from the reviewer, for what specifically needs to be removed or cut down from both the results or discussion and which sentences they feel are too long. As authors of the manuscript, this can sometimes be difficult to see, so some insight would really be appreciated. 

Round 3

Reviewer 2 Report

The authors did not follow the suggestion as regards the shorting-cut of some parts because they were convinced of the importance of the whole text. However, the manuscript is too long and wide. My suggestion about the possibility to improve the manuscript is to divide it into two papers: parts 1 and 2,  where the conclusions are separated.

Author Response

Please find attached a shortened version of the manuscript that has reduced redundancies and removed unnecessary details. Within the word document, track changes highlight what edits have been made to make it clear what actual edits have been made. We have been able to remove 66 lines of text from the manuscript.
